# Learn, Unlearn and Relearn:
# An Online Learning Paradigm for Deep Neural Networks

**Vijaya Raghavan T. Ramkumar**[1]**, Elahe Arani**[‡1,2]**, Bahram Zonooz**[‡1,2]
*vijaya.ramkumar@navinfo.eu, {e.arani, bahram.zonooz}@gmail.com*
[1] *Advanced Research Lab, NavInfo Europe, Eindhoven, The Netherlands*
[2] *Department of Mathematics and Computer Science, Eindhoven University of Technology, The Netherlands*
[‡] *Contributed equally.*

**Reviewed on OpenReview:** *https://openreview.net/forum?id=WN1O2MJDST*

## Abstract

Deep neural networks (DNNs) are often trained on the premise that the complete training data set is provided ahead of time. However, in real-world scenarios, data often arrive in chunks over time. This leads to important considerations about the optimal strategy for training DNNs, such as whether to fine-tune them with each chunk of incoming data (warm-start) or to retrain them from scratch with the entire corpus of data whenever a new chunk is available. While employing the latter for training can be resource-intensive, recent work has pointed out the lack of generalization in warm-start models. Therefore, to strike a balance between efficiency and generalization, we introduce *Learn, Unlearn, and Relearn (LURE)* an online learning paradigm for DNNs. LURE interchanges between the unlearning phase, which selectively forgets the undesirable information in the model through weight reinitialization in a data-dependent manner, and the relearning phase, which emphasizes learning on generalizable features. We show that our training paradigm provides consistent performance gains across datasets in both classification and few-shot settings. We further show that it leads to more robust and well-calibrated models.[1]

## 1 Introduction

> "*A little learning is a dangerous thing.*" -Alexander Pope

In recent years, supervised learning has achieved human-level performance in many computer vision tasks in which the learner is trained in an offline learning environment with a fixed set of training data. However, DNNs deployed in the real world are expected to work in an environment where the data arrive in a sequence of large chunks (mega-batches). Several learning paradigms have been proposed to learn from a stream of data, including, but not limited to, continual learning (Van de Ven & Tolias, 2019; Thrun, 1995), active online learning (Settles, 2009), and anytime learning (Grefenstette & Ramsey, 1992; Caccia et al., 2022). Although previous efforts established a solid theoretical foundation, certain subtle issues make it inapplicable to the practical environment.

For an online learning system, it is important that the learner produces high accuracy and generalizes well at any point in time while using limited computational resources (Caccia et al., 2022). Recent research in online learning, however, has shown that training from a previously trained model (warm-start rather than fresh initialization) hinders its ability to adapt to new input (Achille et al., 2018), thus incapacitating the generalization of DNNs (Ash & Adams, 2020; Caccia et al., 2022). These implications of warm-starting have also been observed in online active learning (Huang, 2021; Sener & Savarese, 2017; Ash et al., 2019), where they mitigate it by retraining from scratch after every selection. However, training DNNs from scratch every time new data arrives is resource intensive, and the lack of generalization with warm-starting undermines the

---

[1]The official code is available at: `https://github.com/NeurAI-Lab/LURE`

benefits of training with learned features (Ash & Adams, 2020). Thus, the failure of current online learning systems to generalize across data streams without bartering previous computation presents a striking lacuna for the large-scale deployment of machine learning systems.

Humans, on the other hand, learn in succession over their lifespan and readily generalize by applying prior knowledge to novel situations and stimuli without the need to learn from scratch. This in the brain is facilitated by a complex set of neurophysiological processes (Goyal & Bengio, 2020). One of such glaring aspects of the brain that allows humans to generalize better is the inherent process of active forgetting (Hardt et al., 2013; Davis & Zhong, 2017). It plays an active role in the regulation of the learning process to achieve better generalizability in the real world. The growing evidence in neuroscience and cognitive psychology (Gravitz, 2019; Izawa et al., 2019) suggests that the brain actively forgets through the selective extinction of neurons, which shapes the learning-memory process and, therefore, prevents humans from overfitting to experiences (Shuai et al., 2010). Thus, emulating this aspect of selective forgetting might hold the key to improving generalization in DNNs.

Therefore, we propose a general online learning paradigm, which we refer to as *Learn, Unlearn, and RElearn (LURE)*, to address the problem of generalization of parameterized networks. For simplicity, we focus mainly on an online learning scenario in which models achieve good performance at any point in time, termed Anytime Learning (Caccia et al., 2022). We consciously simulate the process of selective forgetting (unlearning) in the DNNs by re-randomizing a subset of weights before training on the new samples. With extensive experiments on multiple datasets, we show that our proposed training paradigm boosts the performance and generalization of the models to a greater extent. Compared to standard online training, LURE significantly improves the robustness of DNNs in tackling more challenging real-world scenarios. Our main contributions are as follows:

- "Learn, Unlearn, and Relearn" (LURE), an online training paradigm to improve the performance and generalization of DNNs through the lens of selective forgetting.

- We demonstrate the efficacy of LURE in multiple convolutional architectures across different datasets in online learning and a few-shot classification scenario.

- LURE exhibits robustness in solving more common challenges in real-world problems, including learning with noisy labels, natural corruption, and adversarial attacks.

- LURE is robust to changes in hyperparameters and leads to well-calibrated models.

## 2   Related Work

Lifelong learning (Thrun, 1995) and anytime learning (Grefenstette & Ramsey, 1992) have gained increasing attention from the deep learning community due to its relevance in practical settings. Recent research on online learning, (Caccia et al., 2022; Ash & Adams, 2020), has noticed a lack of generalization in DNNs when trained in online settings. Ash & Adams (2020) points out that if a model is finetuned from a pre-trained model (a "warm-start"), the resulting new model performs worse than a model trained from scratch (a "cold-start") even though the new data is sampled from the same distribution as the previously trained data. Thus, the lack of generalization in DNN renders them inapplicable to real-world scenarios.

Recently, several weight reinitialization methods (Taha et al., 2021; Li et al., 2020; Alabdulmohsin et al., 2021; Ash & Adams, 2020; Zhou et al., 2022) have been proposed to improve the generalization performance of DNNs by partially or fully refining the learned solution. Zhou et al. (2022) propose a forget and relearn hypothesis to unify disparate existing iterative algorithms under the lens of forgetting. Their approach is based on the consideration that early layers learn generalized representation, whereas later layers memorize. Therefore, they reinitialize and retrain the later layers of the model repeatedly, thereby erasing the information pertaining to the memorized difficult examples. Similarly, Ash & Adams (2020) propose a method to improve generalization by shrinking the magnitude of the weights and perturbing it by injecting small noise. However, these weight reinitialization methods have architecture-specific assumptions independent of the data and are handled based on the assumed properties that are inherent to the model and its learning. These methods lack a priori knowledge of where and what features, layers, etc. should be reinitialized in

the general case. Therefore, we propose a online training paradigm, to improve the generalization of DNNs through the lens of active forgetting.

## 3 Method

We propose *"Learn, Unlearn, and Relearn" (LURE)*, a training paradigm for learning from a sequence of data, which alternately interchanges the unlearning (selective forgetting) and relearning steps. Our proposed online training paradigm consists of three steps: a) learn, b) unlearn, and c) relearn. Our proposed approach is illustrated in Figure 1 and is detailed in the Appendix (Algorithm 1).

**Learn.** We define the Anytime Learning at Macroscale (ALMA) learning environment as envisioned in Caccia et al. (2022) where the authors focus on real-world settings. ALMA is a new sub-paradigm of learning from sequential data inspired by anytime learning (Grefenstette & Ramsey, 1992) and transfer learning (Pan & Yang, 2009). Data is provided to the learner in the form of a stream $SB$ consisting of $t$ consecutive batches of samples. Therefore, we also focus on the general classification problem, where data are sampled from an underlying data distribution $\mathcal{D}_{x,y}$ with input $x \in \mathbb{R}^{\mathcal{D}}$ and label $y \in \{1, ..., C\}$.

Let $\mathcal{M}_i$ be a collection of $N \gg 0$ in-distribution samples randomly selected from $\mathcal{D}_{x,y}$, for $i \in \{1, ..., t\}$. The stream is then defined as the ordered sequence $S_B = \{\mathcal{M}_1, ..., \mathcal{M}_t\}$. We refer to each dataset $\mathcal{M}_i$ as a mega-batch, as it is composed of a large number of samples. Consider a model $f_\theta : \mathbb{R}^{\mathcal{D}} \to \{1, ..., C\}$ updates its parameters by processing a mini-batch of $n \ll N$ examples at the time of each mega-batch $\mathcal{M}_i$ in such a way as to minimize its objective function. Since the data are passed as a stream, the model does not have access to the future mega-batches and is limited to one pass through the entire stream. However, the model might make several passes over the current and some previous mega-batches depending on the available computational budget. In ALMA, it is assumed that the rate at which mega-batches arrive is slower than the training time of the model on each mega-batch, and therefore the model can iterate over the mega-batches at its disposal based on its discretion to maximize performance, resulting in an overall data distribution that is not i.i.d. by the end of the stream. This implies a trade-off between effectively generalizing and learning from the current data at each mega-batch. Therefore, in such settings, we train the randomly initialized network $f_{\theta_{Reinit}}$ on a mega-batch $\mathcal{M}_t$ belonging to the data stream $\mathcal{D}_{x,y}$ for $e$ epochs until convergence. The loss function employed for learning is defined as follows:

$$L_\mathcal{T} = \sum_{i=1}^{t} \mathop{\mathbb{E}}_{(x,y)\sim\mathcal{M}_i} \left[ \mathcal{L}_{ce} \left( \sigma \left( f_\theta \left( x \right) \right), y \right) \right], \tag{1}$$

where $\mathcal{L}_{ce}$ is a cross-entropy loss, $t$ is the number of mega-batch sequences, and $\sigma$ is the softmax function.

**Unlearn.** Motivated by the symbiotic link between generalization and active forgetting in biological neural networks (Gravitz, 2019; Davis & Zhong, 2017), we introduce an unlearning step in which the network selectively forgets the connections that are less relevant for the current mega-batch and retains those that are specific for the current mega-batch. We quantify the sensitivity (importance) of each connection in the network in a data-dependent manner to identify task-specific connections. We employ SNIP (Lee et al., 2018), which harnesses the sensitivity of the connection by decoupling the weight from the loss function to find relevant connections. To determine the sensitivity of the connection, we sample a small subset of data from the current mega-batch ($\pi_i \subset \mathcal{M}_i : |\pi_i| = \alpha \times |\mathcal{M}_i|$, where $\alpha$ is the percentage of data to be used as a subset). In our case, following Misra et al. (2022), we use $\alpha = 0.2$, however, we can also use samples as low as 128 (Lee et al., 2018) to estimate the connection sensitivity. We define a connection sensitivity mask $\mathbf{M} \in \{0, 1\}^{|\theta|}$ that is proportional to the number of parameters (m) in the network. Then we apply a sparsity constraint $k$ that specifies the percentage of parameters that must be retained. We compute the connection sensitivity as follows:

$$g_j(\theta; \pi) = \lim_{\delta \to 0} \frac{\mathcal{L}_{ce}(\mathbf{M} \odot \theta; \pi) - \mathcal{L}_{ce}\left((\mathbf{M} - \delta\mathbf{e}_j) \odot \theta; \pi\right)}{\delta} \Bigg|_{\mathbf{M}=1} \tag{2}$$

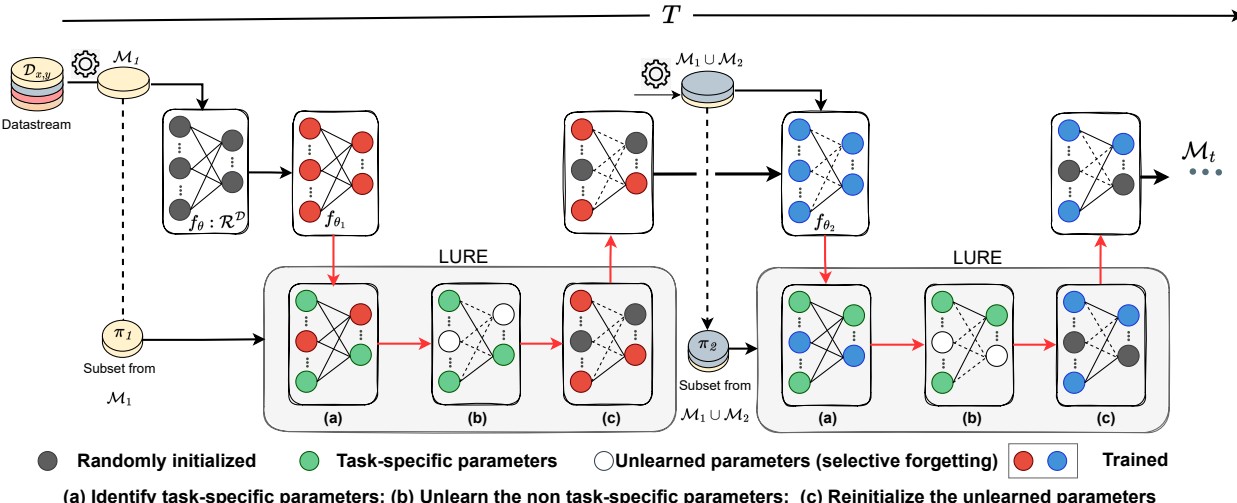

Figure 1: Schematics of the proposed *LURE* framework. LURE alternates between the unlearning phase, which selectively forgets the undesirable information in the model through weight reinitialization, and the relearning phase, which emphasizes learning generalizable features.

where $j$ corresponds to the parameter index and $e_j$ is the mask vector of the index j where the magnitude of the derivatives is then used to calculate the saliency criteria $(s_j)$:

$$s_j = \frac{|g_j(\theta; \pi)|}{\sum_{k=1}^{m} |g_k(\theta; \pi)|}.$$

Following the saliency computation, the connection sensitivity mask is set to only retain the top-k task-specific connections based on the sparsity constraint $k$ which is given as follows:

$$\mathbf{M}_j = \mathbb{1}\left[s_j - \tilde{s}_\kappa \geq 0\right], \quad \forall j \in \{1 \dots m\},$$

where $\tilde{s}_k$ is the $k^{\text{th}}$ largest element in the saliency vector $s$ and $\mathbb{1}[.]$ is the indicator function.

Then, based on the saliencies pertaining to the connection sensitivity, we retain the top-k important connections and unlearn the parameters that are not important for the current data. Thus, we induce active forgetting through reinitialization of the connections that are less desirable for the current mega-batch. Finally, the network parameters pertaining to unlearned connections are reinitialized to random values:

$$\theta_{new}^j = \begin{cases} \theta^j & \text{if } \mathbf{M}_j = 1 \\ \theta_{Reinit} & \text{Otherwise} \end{cases} \tag{3}$$

where $\theta$ are the weight parameters for the previous mega-batch and $\theta_{Reinit}$ corresponds to the random initialized value sampled from a uniform distribution. The network with the new parameters, $\theta_{new}$, is then trained on a new consecutive mega-batch.

**Relearn.** In this stage, the network with the new initialization $(f_{\theta_{new}})$ is updated with the new incoming data $\mathcal{M}_{i+1}$ (in case of no-replay), or with the joint of all the seen data $\mathcal{M}_{i+1} \cup \mathcal{M}_i$ (in case of full-replay) for the $e$ epochs, where $e$ is kept the same for each iteration. The network is trained with the loss function shown in Equation 1. The unlearn and relearn phases are alternately repeated after the completion of each mega-batch training. Thus, by alternating between unlearning and relearning, we favor the preservation of the task-specific connections that can guide the network towards those desirable traits that efficiently improve performance and generalization.

## 4 Experimental set-up

We evaluate our LURE framework on CIFAR-10 (Krizhevsky et al., 2009), CIFAR-100 (Krizhevsky et al., 2009), and Restricted Imagenet (balanced) (Ilyas et al., 2019; Tsipras et al., 2018).

**Anytime learning settings.** Following Caccia et al. (2022), we create the standard for ALMA evaluation using the datasets mentioned above: (1) The training set is randomly divided into $|S_B|$ mega-batches, each containing an equal number of training instances. For CIFAR10 and CIFAR100, we keep $|S_B| = 8$, but for Restricted ImageNet, we keep $|S_B| = 3$. However, we also perform long sequence experiments on CIFAR10 with $|S_B| = 25, 50, 100$ to analyze the effectiveness of the proposed approach. (2) We use 10% of the data from each mega-batch as a mega-batch validation set. Following Caccia et al. (2022), we train each mega-batches for 50 epochs. A test set is employed to evaluate the model's performance as it observes the data. Note that this is not used for validation purposes; it is solely used for final reporting. For implementation details and hyperparameters, please refer to Appendix Section A.5.

**Baselines.** We benchmark our proposed framework against (1) Baseline (BL), a warm-start model, which is continuously trained from the checkpoint of the previously trained model in ALMA settings (without reinitialization) as proposed by Caccia et al. (2022). (2) Cold-start model (Random init.) is trained completely from scratch with each advent of the incoming data. (3) RIFLE (Li et al., 2020), where the fully connected layer is reinitialized and retrained during transfer learning. (4) Shrink and Perturb (S&P) (Ash & Adams, 2020) which is proposed for online learning, and (5) Later-Layer forgetting (LLF)(Zhou et al., 2022) which is proposed to improve generalization in the small data regime.

**Metrics.** For a thorough evaluation, we use CER along with the test accuracy and the generalization gap.

- **Cumulative Error Rate (CER):** This can be defined as follows:

$$CER = \sum_{i=1}^{\mathcal{M}_t} \sum_{j=1}^{|\mathcal{T}_{x,y}|} \mathbb{1}\left(f_\theta^i(x_j) \neq y_j\right), \tag{4}$$

  where $\mathcal{T}_{x,y}$ represents the held-out test set, $f_\theta^i$ is trained on $\mathcal{M}_i$, $y$ is the ground truth label. A model must have a lower CER at each mega-batch of training with a data stream in order to be an effective anytime learner.

- **Generalization gap:** We use the standard generalization gap as a measure to understand whether the model is overfitting or underfitting at anytime learning which is given by the difference between training and validation accuracy.

## 5 Results

### 5.1 Analysis of Short Sequence

Table 1 shows the results of the ResNet18 model training on multiple datasets with and without different forms of reinitialization. All experiments on CIFAR10 and CIFAR100 were carried out using full replay $(S_B = \bigcup_{i=1}^{8} \mathcal{M}_i)$ for a total of 8 mega-batches with each mega-batch containing 6250 samples, while the experiments on Restricted ImageNet are performed for $|S_B| = 3$ mega-batches. Our observations from Table 1 are as follows: (1) Reinitialization-based training for online learning improves test accuracy, CER, and generalization to a greater extent consistently on the three datasets compared to warm-start (BL) and cold-start (Random Init.) models. (2) LURE outperforms the baseline by 4.8%, 6.64% and 4.13% on CIFAR10, CIFAR100, and Restricted ImageNet respectively, and shows the strongest performance on all datasets compared to the other reinitialization methods. (3) Online training using LURE results in the lowest CER and generalization gap compared to other methods, improving the model's anytime learning capabilities. Thus, selective forgetting undesirable information through weight reinitialization brings discernable benefits to the model in online settings.

Table 1: Evaluation of the model (Resnet18) trained with various reinitialization methods in ALMA settings. CIFAR10 and CIFAR100 were trained in a sequence of $|S_B| = 8$, while restricted ImageNet was trained with $|S_B| = 3$ mega-batches.

| Datasets | Methods | Test Accuracy ($\uparrow$) | CER ($\downarrow$) | Generalization Gap ($\downarrow$) |
|---|---|---|---|---|
| CIFAR10 | BL | 89.47 $_{\pm 0.51}$ | 11760 | 8.98 |
| | Random init. | 92.22 $_{\pm 0.57}$ | 12323 | 7.56 |
| | RIFLE | 90.08 $_{\pm 0.37}$ | 11844 | 6.98 |
| | LLF | 91.43 $_{\pm 0.70}$ | 10103 | 7.22 |
| | S&P | 91.76 $_{\pm 0.26}$ | 10206 | 7.72 |
| | LURE | **93.32** $_{\pm 0.58}$ | **9622** | **6.60** |
| CIFAR100 | BL | 62.96 $_{\pm 0.53}$ | 39010 | 31.54 |
| | Random init. | 64.53 $_{\pm 0.51}$ | 37253 | 27.23 |
| | RIFLE | 61.38 $_{\pm 0.26}$ | 39302 | 27.81 |
| | LLF | 67.04 $_{\pm 0.68}$ | 34575 | **21.23** |
| | S&P | 64.48 $_{\pm 0.11}$ | 36303 | 28.58 |
| | LURE | **69.60** $_{\pm 0.82}$ | **33037** | 22.37 |
| Restricted ImageNet | BL | 81.39 $_{\pm 0.48}$ | 2967 | 4.90 |
| | Random init. | 82.87 $_{\pm 0.48}$ | 2852 | 4.88 |
| | RIFLE | 82.05 $_{\pm 0.22}$ | 2722 | **4.63** |
| | LLF | 82.10 $_{\pm 0.85}$ | 2854 | 4.88 |
| | S&P | 80.80 $_{\pm 0.39}$ | 2996 | 5.10 |
| | LURE | **85.52** $_{\pm 0.22}$ | **2699** | 4.84 |

## 5.2 Analysis of Short Sequence with Buffered/No Replay

We investigate a real-world scenario in which access to the entire dataset used to train the model is restricted due to data privacy or memory restrictions. Tables 2 and 3 show the results of training the ResNet18 model on multiple datasets with buffered replay (buffer size =187) and without replay, respectively. All experiments on CIFAR10 and CIFAR100 were carried out for a total of 8 mega-batches with each mega-batch containing 6250 samples, while the experiments on Restricted ImageNet are performed for $|S_B| = 3$ mega-batches. LURE with and without buffered replay consistently outperforms baselines and other methods across all datasets. For example, LURE with buffered replay improves performance by 1.7%, 9.4%, and 2.5% over standard training (BL) on CIFAR10, CIFAR100, and R-ImageNet, respectively. Moreover, we also observe that the performance of the cold-start models (Random Init.) is below par in low buffer and no-reply settings compared to full-replay setting (Table 1) as the model does not have access to past knowledge or data. In the challenging case of no-replay settings, our proposed method has a profound impact on standard training compared to other reinitialization methods. This demonstrates that the observed benefits of LURE are not limited to replay-based methods alone. Thus, unlearning and relearning at each mega-batch of training help to learn more generalizable representation in different replay scenarios.

## 5.3 Analysis of Moderate and Long Sequence ($|S_B| = 25, 50, 100$)

Computational systems deployed in the real world are often exposed to longer mega-batch sequences of data and need to be updated frequently. Therefore, it is quintessential for the online model to perform well under longer mega-batch sequences. Table 4 shows the results of the ResNet18 model training on the CIFAR10 dataset. All experiments were carried out using full replay for a longer sequence of mega-batches 25, 50, and 100. We observe that LURE consistently outperforms the baseline and other methods across varying sequences of mega-batches. As the sequence of mega-batches increases, the number of samples available per mega-batch reduces drastically. Similar to Caccia et al. (2022), we observe that regularly updating the model on fewer samples significantly exacerbates CER, resulting in poor Anytime learning performance. Therefore, long-sequence online learning with reinitialization, especially LURE, reduces the CER and the generalization

Table 2: Evaluation of the model (ResNet18) trained *with buffered replay* (buffer size=186) in ALMA settings. CIFAR10 and CIFAR100 were trained in a sequence of $|S_B| = 8$ mega-batch, while restricted ImageNet was trained for $|S_B| = 3$.

| Datasets | Methods | Test Accuracy (↑) | CER (↓) | Generalization Gap (↓) |
|---|---|---|---|---|
| CIFAR10 | BL | 88.01 ±0.93 | 15784 | 13.01 |
| | Random init. | 75.26 ±0.67 | 22276 | 17.79 |
| | RIFLE | 85.58 ±0.55 | 14748 | 13.50 |
| | LLF | 87.51 ±0.82 | 15045 | 13.22 |
| | S&P | 84.62 ±0.74 | 15549 | 14.72 |
| | LURE | **89.8** ±0.58 | **13629** | **11.92** |
| CIFAR100 | BL | 62.68 ±0.74 | 39178 | 31.33 |
| | Random init. | 58.36 ±0.42 | 40921 | 27.14 |
| | RIFLE | 60.44 ±0.29 | 40040 | 28.19 |
| | LLF | 67.29 ±0.58 | 34416 | **24.86** |
| | S&P | 65.07 ±0.60 | 36040 | 27.79 |
| | LURE | **72.04** ±0.35 | **33037** | 26.59 |
| Restricted ImageNet | BL | 75.97 ±0.55 | 3284 | 8.72 |
| | Random init. | 61.47 ±0.69 | 4023 | **5.41** |
| | RIFLE | 77.90 ±0.48 | 3324 | 8.42 |
| | LLF | 76.40 ±0.65 | **3245** | 9.07 |
| | S&P | 74.75 ±0.38 | 3857 | 8.20 |
| | LURE | **78.58** ±0.49 | 3376 | 7.94 |

gap to a greater extent compared to standard training, thus enriching the predictive capabilities of the model at any given time.

### 5.4 Analysis of Few-Shot Experiments on Restricted ImageNet

For many real-world classification problems, machine learning models deployed often need to be updated on labeled data that are scarce and may not be initially available for training. It is possible for new sets of labeled data to become available gradually as they are labeled. Therefore, it is important for the system to function properly in such online few-shot settings. These experiments are performed on the Restricted ImageNet dataset and quantitatively evaluated with the same metric described in Section 4. We use the same hyperparameters (epochs, lr scheduler), backbone architectures (ResNet50) and optimizer as used in the experiments in ALMA settings (see Appendix Section A.5). However, following Misra et al. (2022), we limit the availability of samples pertaining to the classes to 270. Note that we do not use any techniques proposed in the few-shot literature to boost performance. Table 5 shows the results of the methods in a few-shot settings where we vary the sequence of mega-batches. We observe that reinitialization-based training improves performance and generalization over the baseline, even in challenging few-shot classification. For a mega-batch sequence of 70, LURE outperforms baseline, LLF, and S&P by a relative improvement of 4.11%, 2.01%, and 3.15%, respectively, while for a sequence of 30, it is on par with LLF in terms of accuracy. In addition, LURE outperforms both the important baselines (BL and Rand. Init.) comfortably. In both settings, LURE achieves the lowest CER and the generalization gap, demonstrating the superiority of our proposed approach.

### 5.5 Analysis on Various Architectures

Here, we examine the versatility of our proposed LURE framework for multiple architectures on the CIFAR10 dataset. We consider ResNet18 (He et al., 2016), ResNet50 (He et al., 2016), wider-Resnet50-2 (Zagoruyko & Komodakis, 2016), VGG16 (Simonyan & Zisserman, 2014). We chose these models explicitly because of their widespread popularity in common computer vision tasks and the breadth of research done on them for

Table 3: Evaluation of the model (ResNet18) trained *without replay* in ALMA settings. CIFAR10 and CIFAR100 were trained in a sequence of $|S_B| = 8$ mega-batch, while restricted ImageNet was trained for $|S_B| = 3$.

| Datasets | Methods | Test Accuracy (↑) | CER (↓) | Generalization Gap (↓) |
|---|---|---|---|---|
| CIFAR10 | BL | $80.86_{\pm1.01}$ | 16789 | 16.85 |
| | Random init. | $73.56_{\pm0.45}$ | 22731 | 17.17 |
| | RIFLE | $86.35_{\pm0.14}$ | 14536 | 15.53 |
| | S&P | $81.92_{\pm0.62}$ | 16133 | 16.36 |
| | LLF | $86.11_{\pm0.59}$ | 15294 | 14.66 |
| | LURE | $\mathbf{88.96}_{\pm0.24}$ | **13953** | **12.14** |
| CIFAR100 | BL | $48.79_{\pm0.75}$ | 44128 | 49.95 |
| | Random init. | $35.50_{\pm0.66}$ | 55221 | 67.03 |
| | RIFLE | $45.65_{\pm0.42}$ | 44113 | 51.46 |
| | LLF | $50.38_{\pm0.54}$ | 44816 | 50.80 |
| | S&P | $49.26_{\pm0.59}$ | 44581 | 51.45 |
| | LURE | $\mathbf{55.37}_{\pm0.63}$ | **43348** | **46.69** |
| Restricted ImageNet | BL | $73.36_{\pm0.88}$ | 3450 | 9.68 |
| | Random init. | $60.87_{\pm0.61}$ | 4179 | **7.18** |
| | RIFLE | $74.87_{\pm0.24}$ | 3375 | 8.36 |
| | LLF | $76.40_{\pm0.65}$ | 3515 | 8.15 |
| | S&P | $73.36_{\pm0.78}$ | 3464 | 8.13 |
| | LURE | $\mathbf{77.97}_{\pm0.43}$ | **3367** | 7.90 |

Table 4: Evaluation of the model (ResNet18) on CIFAR10 for longer sequences of mega-batches.

| # Mega-batches | Methods | Test Accuracy (↑) | CER (↓) | Generalization Gap (↓) |
|---|---|---|---|---|
| 25 | BL | $89.94_{\pm0.54}$ | 43029 | 6.80 |
| | Random init. | $89.89_{\pm0.36}$ | 43529 | **4.86** |
| | RIFLE | $90.35_{\pm0.54}$ | **41308** | 5.55 |
| | LLF | $89.80_{\pm0.62}$ | 42961 | 6.52 |
| | S&P | $88.37_{\pm0.26}$ | 43578 | 5.94 |
| | LURE | $\mathbf{90.55}_{\pm0.34}$ | 42790 | 6.79 |
| 50 | BL | $89.12_{\pm0.61}$ | 87843 | 6.02 |
| | Random init. | $89.62_{\pm0.42}$ | 91040 | 6.02 |
| | RIFLE | $89.25_{\pm0.53}$ | 91426 | 6.16 |
| | LLF | $90.26_{\pm0.82}$ | 87826 | 5.75 |
| | S&P | $88.32_{\pm0.35}$ | 85798 | 6.11 |
| | LURE | $\mathbf{90.97}_{\pm0.67}$ | **85487** | **5.18** |
| 100 | BL | $89.64_{\pm0.69}$ | 176954 | 6.46 |
| | Random init. | $89.29_{\pm0.39}$ | 188978 | 5.61 |
| | RIFLE | $89.93_{\pm0.27}$ | 170678 | 6.02 |
| | LLF | $89.48_{\pm0.77}$ | 173505 | 7.21 |
| | S&P | $88.01_{\pm0.44}$ | 182294 | **5.56** |
| | LURE | $\mathbf{91.95}_{\pm0.53}$ | **170178** | 5.66 |

different learning paradigms. Table 6 shows the performance and generalization gap of the model trained in different architectures. The experiments are performed with full replay for $|S_B| = 4$. The results demonstrate that LURE significantly outperforms standard training across multiple architectures while the LLF and S&P fail to improve. Therefore, reinitialization of the weight parameter in a data-dependent manner using

Table 5: Few-shot experiments using ResNet50 on Restricted ImageNet.

| # Mega-batches | Methods | Test Accuracy (↑) | CER (↓) | Generalization Gap (↓) |
|---|---|---|---|---|
| 30 | BL | 45.82 $_{\pm 0.55}$ | 73514 | 47.93 |
| | Random init. | 45.79 $_{\pm 0.57}$ | 73497 | 45.77 |
| | RIFLE | 44.58 $_{\pm 0.46}$ | 74071 | 46.30 |
| | LLF | **47.60** $_{\pm 0.61}$ | 72868 | 43.50 |
| | S&P | 45.92 $_{\pm 0.38}$ | 72954 | 45.61 |
| | LURE | 46.97 $_{\pm 0.79}$ | **71542** | **42.56** |
| 70 | BL | 53.43 $_{\pm 0.49}$ | 156332 | 41.17 |
| | Random init. | 54.20 $_{\pm 0.69}$ | 150473 | 38.17 |
| | RIFLE | 53.25 $_{\pm 0.52}$ | 153716 | 36.31 |
| | LLF | 54.53 $_{\pm 0.68}$ | 149493 | 32.57 |
| | S&P | 53.90 $_{\pm 0.24}$ | 150332 | 40.87 |
| | LURE | **55.65** $_{\pm 0.52}$ | **148478** | **28.35** |

Table 6: Evaluation of methods using different architectures on CIFAR10 dataset ($|S_B| = 4$).

| Architechture | Methods | Test Accuracy (↑) | CER (↓) | Generalization Gap (↓) |
|---|---|---|---|---|
| ResNet18 | BL | 89.31 $_{\pm 0.61}$ | 5657 | 7.90 |
| | Random init. | 92.78 $_{\pm 0.31}$ | 5459 | 7.56 |
| | RIFLE | 92.11 $_{\pm 0.38}$ | 5378 | 7.77 |
| | LLF | 91.80 $_{\pm 0.43}$ | 5466 | 7.68 |
| | S&P | 90.50 $_{\pm 0.51}$ | 5667 | 7.80 |
| | LURE | **93.73** $_{\pm 0.35}$ | **4409** | **7.48** |
| ResNet50 | BL | 89.53 $_{\pm 0.58}$ | 6854 | 7.83 |
| | Random init. | 91.87 $_{\pm 0.66}$ | 7539 | 6.78 |
| | RIFLE | 90.51 $_{\pm 0.35}$ | 6740 | 6.95 |
| | LLF | 89.49 $_{\pm 0.45}$ | 6782 | 6.78 |
| | S&P | 89.15 $_{\pm 0.61}$ | 6940 | 6.98 |
| | LURE | **92.75** $_{\pm 0.73}$ | **6682** | **6.62** |
| Wide-ResNet50-2 | BL | 90.10 $_{\pm 0.69}$ | 5978 | 6.84 |
| | Random init. | 92.23 $_{\pm 0.27}$ | 6615 | 6.73 |
| | RIFLE | 89.98 $_{\pm 0.42}$ | 5953 | 6.02 |
| | LLF | 89.72 $_{\pm 0.36}$ | 6000 | **5.70** |
| | S&P | 89.38 $_{\pm 0.61}$ | 6292 | 5.96 |
| | LURE | **93.78** $_{\pm 0.54}$ | **5557** | 5.81 |
| VGG16-BN | BL | 89.32 $_{\pm 0.74}$ | 5650 | 9.62 |
| | Random init. | 91.7 $_{\pm 0.33}$ | 5913 | 7.22 |
| | RIFLE | 88.25 $_{\pm 0.38}$ | 6083 | 6.89 |
| | LLF | 87.85 $_{\pm 0.58}$ | 6124 | **6.17** |
| | S&P | 88.25 $_{\pm 0.86}$ | 5720 | 8.11 |
| | LURE | **92.67** $_{\pm 0.47}$ | **4439** | 8.55 |

connection sensitivity by LURE is more effective to improve generalization in different architectures than reinitialization based on assumed model properties and learning (as done by LLF and S&P).

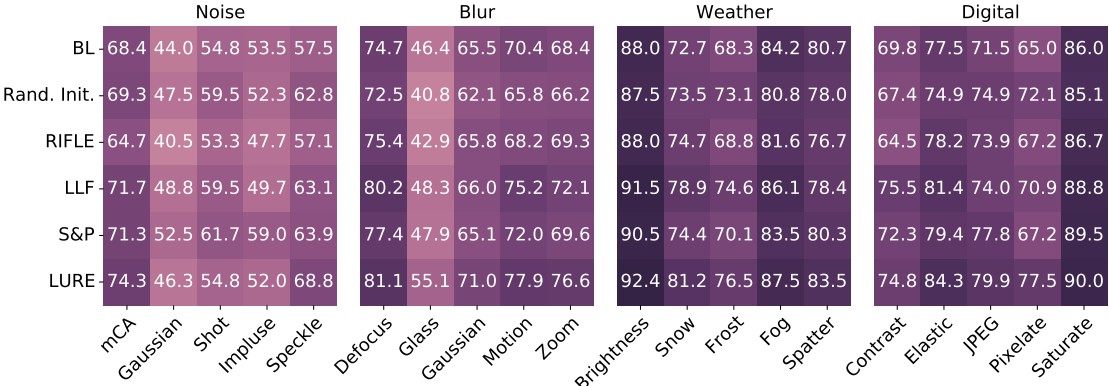

Figure 2: Robustness to natural corruptions on CIFAR10-C (Hendrycks & Dietterich, 2019). LURE is more robust against the majority of corruptions compared to other reinitialization methods.

## 6 Robustness Analyses

### 6.1 Robustness to Natural Corruptions

In practice, DNNs are often deployed in real-world scenarios where they are exposed to constantly changing environments, often influenced by changes in lighting and weather. Therefore, the robustness of DNNs to data distributions that are subject to natural corruption is pertinent. Here, we evaluate the benefit of LURE on robustness to common corruption using CIFAR10-C (Hendrycks & Dietterich, 2019). Models are trained on clean images and tested on CIFAR10-C. Following Hendrycks & Dietterich (2019), we use the mean Corruption Accuracy (mCA) to measure performance under natural corruption. Figure 2 shows the accuracy of the models on 19 different corruptions averaged on five severity levels. Compared to baseline (68%), Random initialization (69.3%), RIFLE (64.7%), LLF (72%), and S&P (71%), LURE (74%) delivers a higher mCA in all types of corruption. Evidently, unlearning and relearning at each mega-batch of training bring discernible benefits in terms of robustness to natural corruptions.

### 6.2 Robustness to Adversarial Attacks

DNNs have been shown to be vulnerable to adversarial attacks in which imperceptible perturbations are added to inputs during inference. The adversarial images are designed to fool the network to make false predictions (Szegedy et al., 2013). We perform a PGD-10 attack (Madry et al., 2017) on models trained on the CIFAR10 dataset with varying attack strengths. As observed in Figure 3(Left), LURE exhibits greater resistance to these attacks of varying strengths. Thus, compared to standard training, training a model in the LURE framework facilitates online learners to learn high-level abstractions that are not sensitive to small perturbations in the data.

### 6.3 Robustness to Noisy Labels

The success of supervised learning often depends on the availability of large amounts of high-quality annotations. However, the availability of high-quality annotated datasets can be extremely expensive and time consuming to collect. Therefore, it is paramount to have robust training on noisy labels, as studies have shown that DNNs can easily memorize samples and are susceptible to noisy labels (Arpit et al., 2017). we train ResNet18 on sequential CIFAR10 with noisy labels for a mega-batch sequence of $|S_B| = 4$ and evaluate the performance on a clean test set. We corrupt every ground truth label with a specified probability (noise rate) by randomly sampling from a uniform distribution over a large number of classes. Figure 3(Right) presents the test accuracy of the reinitialization methods under different percentages of noisy labels. The results show that the reinitialization-based training paradigm is robust to the presence of noisy labels during training compared to the warm-started baseline. Furthermore, LURE consistently outperforms the LLF, RI-

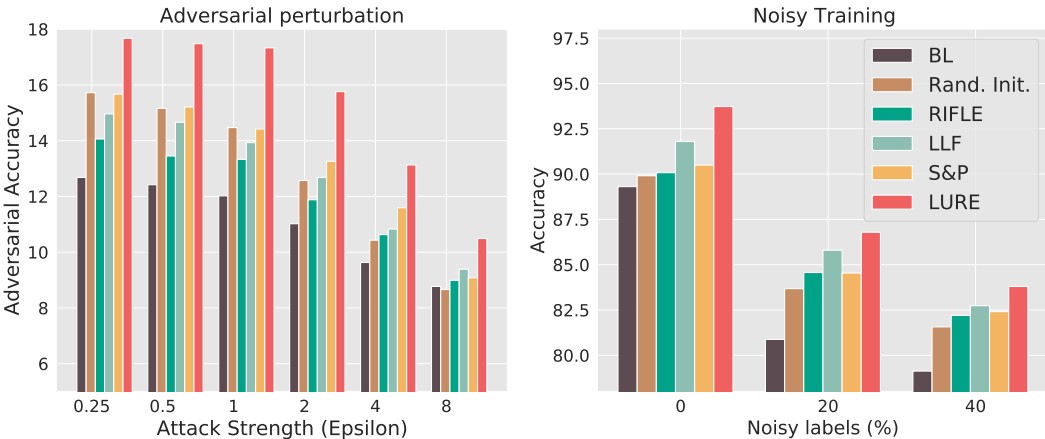

Figure 3: (Left) Robustness to adversarial attacks; (right) Robustness to training under noisy labels. In both robustness analyses, LURE shows a significant performance improvement compared to the baselines considered.

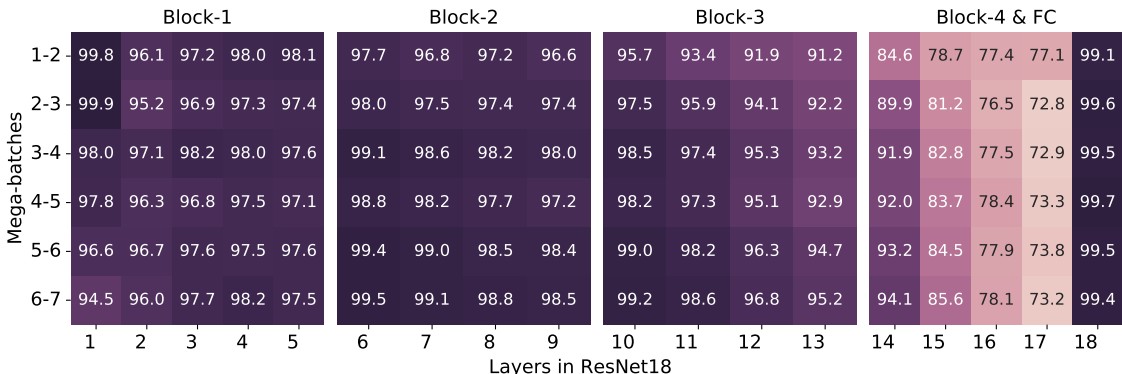

Figure 4: Layer-wise percentage overlap of the retained parameters in consecutive mega-batches in the full-replay scenario. The results for the no-replay and buffered replay scenarios are provided in Figures 9(a,b), respectively.

FLE, and S&P baseline comfortably across different noisy label rates. Thus, reinitializing based on selective forgetting helps to learn a generic representation that is less sensitive to noise in the dataset.

## 6.4 Robustness of Connection Selection across Training Steps

The proposed LURE framework is based on the selection of connections (that is, SNIP (Lee et al., 2018)) to selectively forget the parameters that have the least impact on performance at each mega-batch of training. Therefore, it is important to evaluate the consistency of the connection selection across the sequence of mega-batches. We conduct this analysis on the short sequence Anytime learning scenario (full replay, $((|S_B| = 2))$ on CIFAR10 with ResNet18. We globally retain 80% and reinitialize (unlearn) 20% of the parameters iteratively at the end of each mega-batch of training.

To study the consistency of the retained connections, we save the connection sensitivity mask (M) containing ones and zeros for the parameters retained and to be reinitialized, respectively. Visualizing the connection sensitivity mask can be challenging, as the parameter count in each layer of the backbone is overwhelming. Therefore, we calculate the percentage of overlap of retained parameters between the connection sensitivity mask generated at the end of consecutive mega batch training as a metric to analyze the connection consistency. Figure 4 shows the layer-wise percentage overlap of the retained parameters across consecutive

mega-batches. The overlap percentage of retained connections is quite high in the earlier layers across all mega-batches, while it decreases in the later layers (layer 4 in ResNet) as it learns class-specific information pertaining to the new (unseen) samples. Although the overlap percentage is lower in the later layers, our method selectively unlearns few parameters in the earlier layers as well depending on the incoming data. This flexible nature of our method for unlearning and regulating connections in both the latter and early layers facilitates improved generalization in Anytime setting. This result is consistent not only in the full replay, but also in buffered and no-replay scenarios of anytime learning. The results for the no-replay and buffered replay scenarios are provided in Figures 9(a,b), respectively.

### 6.5 Sensitivity to Hyperparameters

Machine learning systems are often deployed in the real world, where explicitly running a hyperparameter search for each update can be computationally exhaustive. Therefore, similar to Zaidi et al. (2022), we explore the sensitivity of our method to the choice of weight decay and learning rate. Figure 5 shows the test accuracy achieved by changing the learning rate and the weight decay values intended for CIFAR-10 training in ALMA scenarios for the mega-batch sequence of $|S_B| = 4$. Compared to baseline training without reinitialization, LURE is less sensitive to the choice of hyperparameters. A detailed comparison with other methods is provided in Appendix, Figure 8. For example, the performance of normal training decreases to 75% with a learning rate of 0.005 and a weight decay of 0.001, while the performance of LURE remains above 94% throughout. Therefore, LURE can improve generalization in regimes where it is infeasible to perform exhaustive hyperparameter tuning.

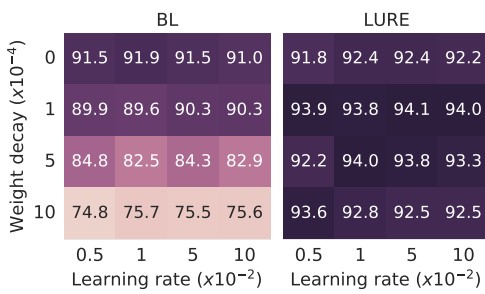

Figure 5: Sensitivity to hyperparameters. LURE is more robust to the changes in weight decay and learning rate to standard online training.

We underline that LURE's effectiveness exceeds that of other reinitialization techniques and that it should be viewed as a general-purpose online training paradigm, as it is more resilient to typical problems found in real-world datasets than the conventional online training method. Extensive experiments on the model characteristic analyses, such as model calibration and convergence to flatter minima, are provided in Appendix Sections A.3 and A.4, respectively. Furthermore, we analyze the robustness of connection selection during training steps and also across experiments trained with different learning rates. Finally, we have added a discussion section comparing LURE with other baselines and the benefits it brings in practical settings where the training budget is limited.

## 7 Conclusion and Future work

We introduce *Learn, Unlearn, and RElearn (LURE)*, an online training paradigm to improve DNN performance and generalization through the lens of selective forgetting. LURE alternates between the unlearning phase, which selectively forgets undesirable information in the model, and the relearning phase, which emphasizes learning generalizable features. Empirical results show that the proposed framework improves performance and generalization across a wide range of architectures and datasets, both online and in challenging few-shot classification. Our framework is robust to learning with noisy labels and adversarial attacks and increases generalization in many real-world scenarios. One advantage of our work is that we have observed distinct empirical tendencies when re-initialization succeeds. In future work, it would be interesting to study the dynamics of reinitialization in other lifelong learning scenarios, such as continual learning, where domain shifts and catastrophic forgetting are more common. Furthermore, studying activation-based connection selection may help us selectively identify and retain the most important weights in online learning settings. Further research in these areas may provide a more in-depth theoretical explanation for why reinitialization succeeds or fails.

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

# A   Appendix

## A.1   Broader Impact and Societal Relevance

We believe that our findings can potentially be harnessed to enhance the test accuracy and robustness of any machine learning system deployed in the real world where the aspect of generalization is crucial, as they are continuously trained on sequential data. For example, consider a large-scale social media website in which users continually upload images and content. To recommend material, filter out inappropriate media, and choose adverts, the organization requires up-to-date prediction models. Every day, millions of fresh data points may arrive, which must be quickly integrated into operational ML pipelines. In this scenario, it is logical to envision having a single model that is regularly updated with the most recent data. Every day, additional training on the model with the updated and larger dataset might be undertaken. In these scenarios, the proposed framework (LURE) can improve the generalization and performance of the model to a greater extent as opposed to new training from the parameters of yesterday's model without reinitialization.

Furthermore, in applications such as autonomous driving and industrial robotics, where the deployed model needs to be frequently updated in order to stay in sync with the surroundings. Using LURE as a training paradigm to update the model can boost performance and generalization in a computationally efficient way, as it provides a better initialization for continuous training compared to warm-starting or updating the model from scratch.

In addition to the above scenarios, our proposed framework can be conceivably harnessed in applications of deep active learning where the goal is to find the most informative data to label with an oracle and incorporate into the training set. However, current active learning frameworks retrain models from scratch after each querying step, which is computationally expensive and partially responsible for deleterious environmental ramifications. The LURE framework allows models to be efficiently updated without sacrificing generalization and performance, thus having a positive impact on society.

## A.2   Ablation studies

To examine the influence of the individual components of our LURE network in ALMA settings, we perform the following ablation study.

**Effect of Ratio of Reinitialized Parameters.** Table 7 shows the effect of varying the number of initialized parameters on the performance and generalization of the model in CIFAR10. We train the model in ALMA settings using the LURE framework by varying different percentages of reinitialized parameters (5%, 10%, 20%, 30%, and 40%). Experiments were carried out using full-replay with ResNet18 for $|S_B| = 8$. The results show that the unlearning of a 5% percentage of parameters has no impact on performance, while the unlearning of more than 30% has less impact on test accuracy. We find that reinitialization 20% of the parameters results in the best performance.

Table 7: Evaluation varying the percentage of reinitialized parameters during training on CIFAR10 dataset using ResNet18.

|      | Reinitialized Params (%) | Test Accuracy (↑) | CER (↓) | Generalization Gap (↓) |
|------|--------------------------|-------------------|---------|------------------------|
|      | 5  | 89.84 | 11955 | 6.83 |
|      | 10 | 91.81 | 11571 | 7.22 |
| LURE | 20 | **94.32** | **9622** | 6.60 |
|      | 30 | 90.73 | 11806 | 7.91 |
|      | 40 | 90.79 | 11559 | **6.57** |

**Effect of Importance Estimation Method.** We investigate the effect of various methods of importance estimation on our proposed training paradigm. For this, we consider Fisher Importance (FIM), weight magnitude, random, and SNIP. Table 8 demonstrates the performance and generalization of the model trained with the LURE framework with different selections to estimate important parameters on CIFAR10 using

Table 8: Evaluation with different importance estimation on CIFAR10 dataset.

|  | Importance criteria | Test Accuracy (↑) | CER (↓) | Generalization Gap (↓) |
|---|---|---|---|---|
| LURE | BL | 89.31 | 5657 | 7.90 |
|  | FIM | 92.73 | **4346** | 8.39 |
|  | Weight Magnitude | 92.18 | 4547 | 8.53 |
|  | SNIP | **93.73** | 4409 | **7.48** |

Table 9: Evaluation with varying the quantity of data for importance estimation on CIFAR10 dataset.

|  | # samples | Test Accuracy (↑) |
|---|---|---|
| LURE | $0.2\,|\mathcal{M}|$ | 93.73 |
|  | 128 | 93.62 |

ResNet18 for $|S_B| = 4$. Unlearning and relearning with SNIP, Fisher information, and weight magnitude results in better performance compared to baseline. This shows that our training paradigm is not only limited to SNIP, but any importance estimation criterion can be used to identify the dataset-specific connections.

**Varying the quantity of data used for Importance estimation** In our experiments, we randomly sampled 20% of the data from each mega-batch and used it to estimate the importance of the parameters before selective forgetting. Here, we analyze the impact of the number of data used to determine the important estimation on the final performance. Similar to Lee et al. (2018), we used as few as 128 samples to estimate the important parameters using SNIP. Table 9 shows that LURE is not sensitive to the variation in the input data used to estimate the importance as the final performance remains unchanged.

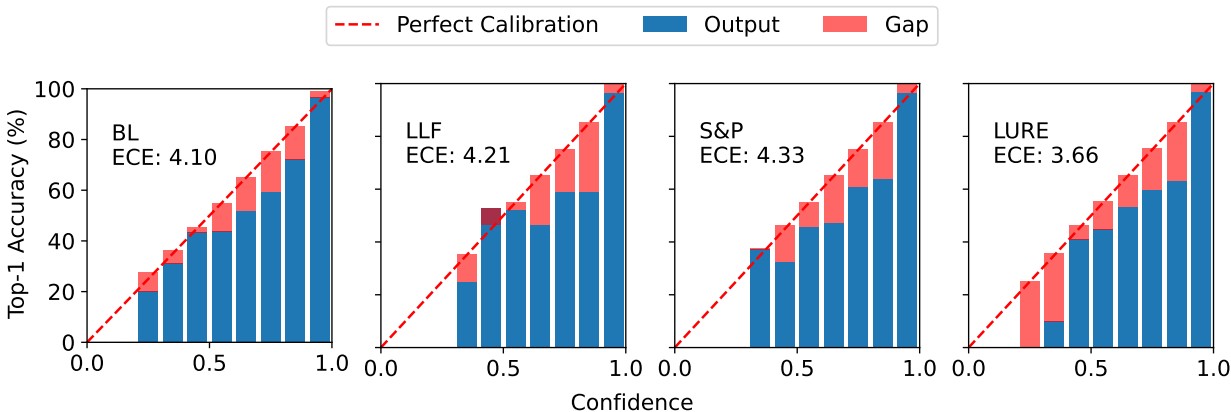

Figure 6: Confidence estimates and the corresponding Expected Calibration Error (ECE) of the CIFAR-10 ALMA trained models. Lower ECE is better. Our method is well calibrated, with confidence estimates closer to perfect calibration compared to BL, LLF, and S&P.

## A.3 Model Calibration

DNNs are often deployed in safety-critical applications, where it is essential to have a model that has a sufficient sense of uncertainty about its predictions. Therefore, we evaluate the calibration of models trained with different reinitialization methods in ALMA settings. The common metric for identifying miscalibration in classification is the Expected Calibration Error (ECE) (Naeini et al., 2015). The ECE measures the discrepancy between absolute accuracy and average confidence as a weighted average. The lower the ECE, the better calibrated the model is. Figure 6 shows the ECE values along with a reliability diagram on CIFAR10 using the calibration library by Kuppers et al. (2020). The result shows that BL, LLF, and S&P

are highly miscalibrated and far more overconfident than the proposed LURE framework. Thus, in addition to improving performance and generalization, online learning using selective forgetting can effectively improve calibration, thus improving reliability in contexts where safety is of absolute importance.

### A.4 Convergence to Flatter Minima

DNNs that converge to flatter minima in a loss landscape have greater adaptability to new tasks without straying too far from the optimal parameters for previous tasks. Furthermore, solutions that reside in flatter minima are more robust because the predictions do not change significantly with minor perturbations. We apply independent Gaussian noise to all parameters of the CIFAR-10 trained model, as described in (Alabdulmohsin et al., 2021). Figure 7 shows that the solution reached by LURE, LLF, and S&P is more robust to model perturbation than standard training. Our method is significantly less sensitive to perturbations than the other methods, and the performance gradually decreases. More specifically, for every amount of noise introduced into the model parameters $\theta$, the change in LURE training accuracy is smaller than in standard training, implying that the solution provided by LURE appears to reside in flatter local minima. We argue that training the model by alternating between learning and unlearning stages leads to a larger valley, which could better explain our model's ability to consolidate generalizable features.

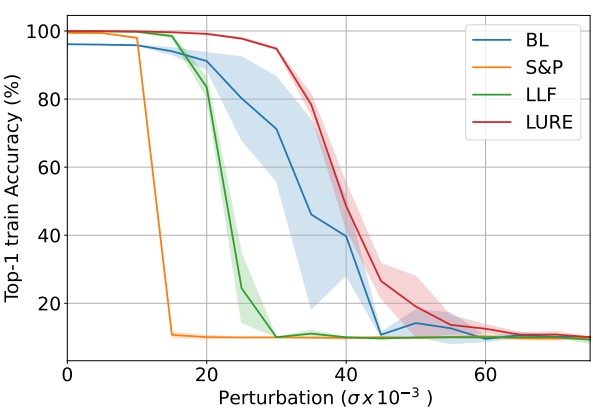

Figure 7: Robustness of the model perturbed by varying degrees of Gaussian noise. Our method is considerably robust to Gaussian perturbations, as the decline in performance is gradual, suggesting convergence to flatter minima.

### A.5 Implementation Details

**Datasets.** We empirically evaluate our proposed framework on three different data sets: (a) CIFAR-10 (Krizhevsky et al., 2009) (b) CIFAR-100 (Krizhevsky et al., 2009) and (c) Restricted Imagenet (balanced) (Ilyas et al., 2019; Tsipras et al., 2018). CIFAR10 and CIFAR100 consist of 50,000 training images and 10,000 test images, each of size $32 \times 32$, divided into 10 and 100 classes, respectively. Restricted ImageNet (balanced) is a subset of the original ImageNet data set (Russakovsky et al., 2015) consisting of 89517 training images and 3450 test images, each of $224 \times 224$ size divided into 14 classes consisting of five subclasses each. For ease of computation, we resize the images to $32 \times 32$ for our experimental settings.

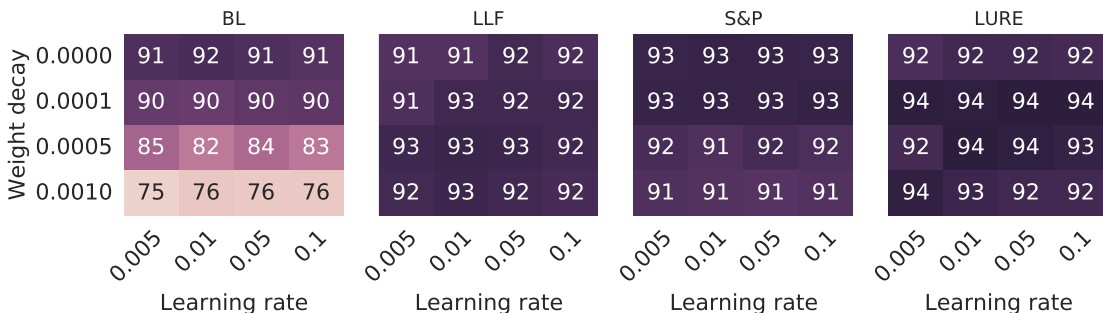

Figure 8: The sensitivity of different reinitialization methods to hyperparameters. The initialization-based training paradigm is more robust against the change in weight decay and the learning rate compared to standard online training.

---

**Algorithm 1** Training LURE in ALMA settings

---

    **input:** Data stream $\mathcal{S}_B = \{\mathcal{M}_1, ..., \mathcal{M}_t\}$, Model $f_\theta^{i=0}$, replay, Sparsity $\alpha$
1:  $i \leftarrow 1$
2: **while** $i \leq |S_B|$ **do**
3:     **if** replay **then**
4:         $\mathcal{M}_i \leftarrow \bigcup_{i=1}^{t} \mathcal{M}_i$
5:     **else**
6:         $\mathcal{M}_i \leftarrow \mathcal{M}_i$
7:     $f_\theta^i \leftarrow f_\theta^{i-1}.train(\mathcal{M}_i)$                                ▷ Training or learning step
8:     $\pi_i \leftarrow 0.2\mathcal{M}_i$
9:     $M \leftarrow$ Importance Estimation$(f_\theta^i, \pi_i, \alpha)$
10:    Retain the task specific weights based on $M$
11:    Randomly reinitialize the task irrelevant parameters in $f_\theta^i$               ▷ Selective forgetting
12:    Model with this new initialization for next $\mathcal{M}_{i+1}$ training

---

**Implementation Details.** The efficacy of our framework is demonstrated in ALMA settings (Caccia et al., 2022). ResNet18 (He et al., 2016) is used as the backbone for most of the experiments on CIFAR10 and CIFAR100, while ResNet50 (He et al., 2016) is used for restricted ImageNet experiments. We initialize the networks randomly and use stochastic gradient descent (SGD) with momentum 0.9 and weight decay 1e-4 to optimize it. We follow the same procedure as that followed by Caccia et al. (2022); Misra et al. (2022). Networks are trained iteratively for $t$ mega-batches with a batch size b = 64 for 50 epochs per mega-batch of training without early stopping. A step learning rate scheduler with an initial learning rate of 0.1 decayed at steps 20 and 40 is employed during the mega-batch training of our method. The standard data augmentation technique, that is, flipping and random cropping, is used. All training settings (lr, b, e) are kept constant throughout the mega-batch training. We randomly divide the data set into mega-batches with an equal number of samples in each mega-batch. For each mega-batch $\mathcal{M}_t$, we divide it into a train set with 90% of the samples and a validation set with the remaining 10% samples. We then randomly sampled 20% of the training data from each mega-batch to build the set $\pi$ used to identify task-specific parameters through SNIP after each mega-batch of training. We use the default parameters for the SNIP algorithm specified above. Finally, we maintain a separate held-out test set, which is used to evaluate the model's performance after training on each mega-batch. Unless specified, we keep the number of mega-batches to 8 for all the experiments. For training S&P, a shrink coefficient of 0.4 and a noise of 0.001 are applied for the weights of the entire network before training on the new mega-batch of data. Similarly, for LLF, we reinitialize blocks 3 and 4 of ResNet (He et al., 2016) before the start of each mega-batch of training whereas for RIFLE we only reinitialize the last fully connected classification layer. For few-shot experiments, we do not consider existing techniques proposed in the few-shot literature. We limit the number of samples from each class to 270, which are sampled randomly in a class-balanced way. Finally, we use the same hyperparameters to perform experiments with different datasets and architectures.

### A.6   Robustness of connection selection to change in learning rate.

Reiterating from Section 6.5 where our LURE framework is less sensitive to the choice of hyperparameters such as learning rate and weight decay. Here, we explore the sensitivity of connection selection to the choice of hyperparameters. Similar to the experiments in Section 6.4, we calculate the layer-wise percentage overlap of retained parameters across different training setups at the end of mega batch training. We compute the overlap between models trained with different learning rates. We keep the seed and other hyperparameters consistent across the experiments. The results in Figure 10 demonstrate that the selection of connections is more robust to the model trained at different learning rates. While most of the connections retained in the early layers remain consistent across training setups, the later layers (layer 4) showed less overlap for the model trained with different learning rates. Nevertheless, our connection selection is robust to the choice of learning rate.

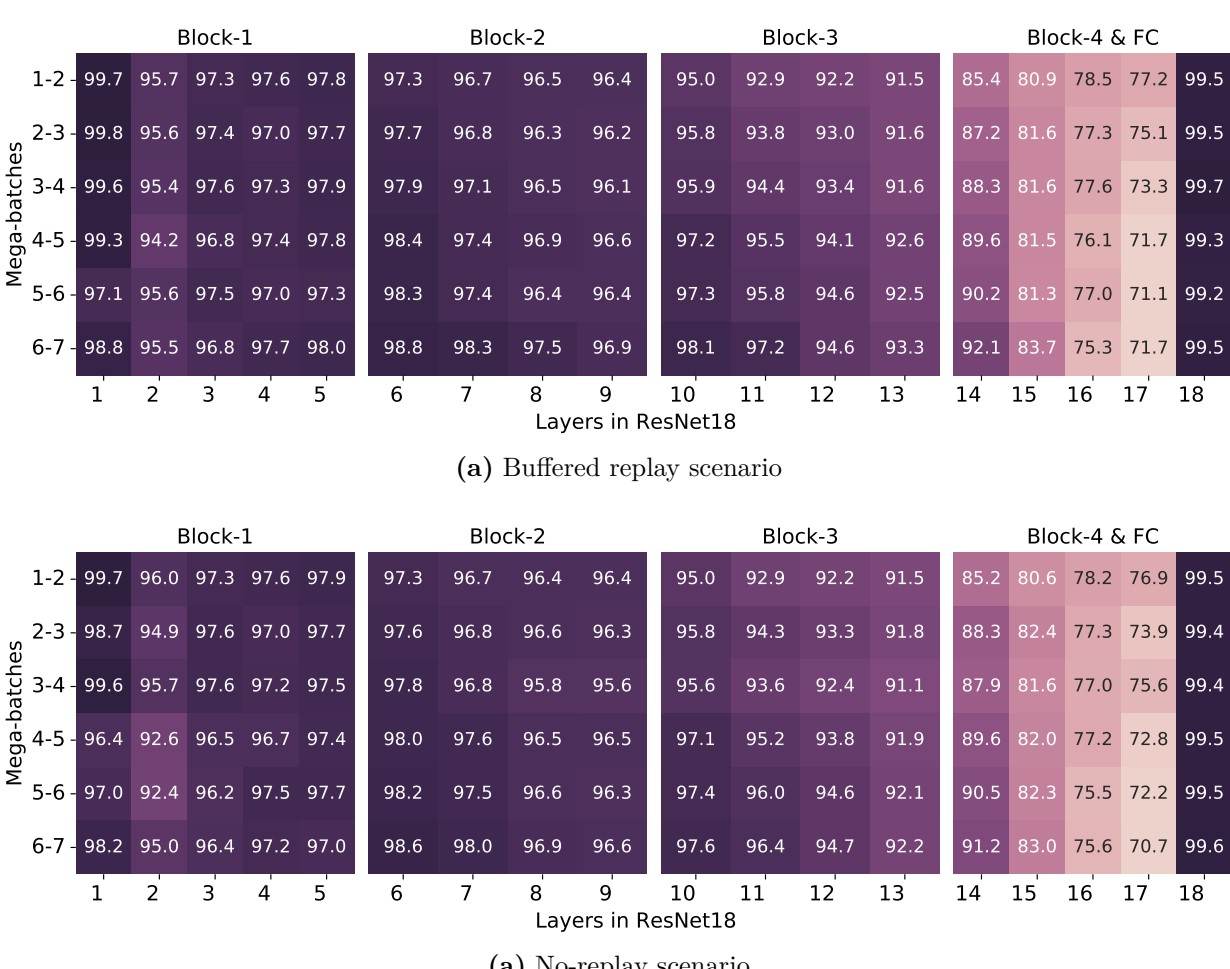

**(a)** Buffered replay scenario

**(a)** No-replay scenario

Figure 9: Layer-wise percentage overlap of the retained parameters across consecutive mega-batches. The percentage of connections overlapped between consecutive mega-batches remains consistent in the early and mid layers (block-1, 2, and 3) whereas it changes in the latter layers as the model is trained on the new incoming data.

### A.7 Evaluating the redundancy of parameters removed during the unlearning phase.

Since our proposed method selectively forgets connections during training, we evaluate the redundancy of the unlearned connection in the model in terms of performance and robustness. To analyze this, we measure the performance and robustness of dense and sparse models before and after selective forgetting. For this, we measure the performance of the ResNet18 model trained on CIFAR10 with $|S_B| = 8$ after the first mega-batch training. The performance is evaluated with the clean test dataset, while generalization is evaluated with the test images subjected to 15 types of natural corruptions. Table 10 shows that the drop in test accuracy between the dense model and the sparse model (containing 20% fewer parameters than the dense model) is just 0.31% which is insignificant. The relative drop in test and robust accuracy with respect to train accuracy is less with the sparse model when compared to the dense model. In addition, in robustness and generalization analysis, the sparse model either outperforms or achieves the same accuracy when compared to the dense model. This shows that the connections that are constantly reinitialized during training contain trivial information that is redundant in fact and adds insignificant value to model training in terms of generalization and performance. Thus, we empirically show that selective forgetting is crucial for retaining previous information while freeing the model's capacity to learn incoming data, thereby improving generalization in anytime learning.

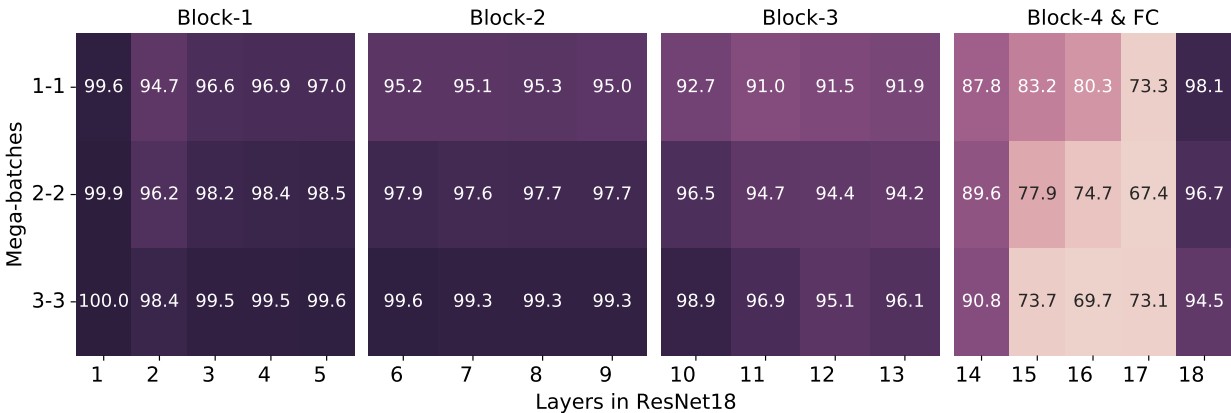

Figure 10: Layer-wise overlap percentage of the retained parameters across training runs with different learning rates.

Table 10: Evaluating the redundancy of the parameters removed during the unlearning phase. The relative improvement between the train accuracy and test/robust accuracy is shown in brackets.

|  | # Parameters | Train Accuracy (↑) | Test Accuracy (↑) | Robust Accuracy (↑) |
|---|---|---|---|---|
| Full model | 11173962 | 75.54 | 71.78 (0.95%) | 55.80 (0.74%) |
| Sparse model | 8939169 | 74.04 | 71.47 (0.96%) | 56.20 (0.76%) |

## A.8 Comparison of LURE with warm-start and cold-start training.

Figure 11 shows a comparison between ResNet18 trained using warm-start, random initialization, and LURE on CIFAR-10 for a mega-batch size of $((|S_B| = 2)$. For the first 50 epochs, the models are trained with 50% of data. Then, it is trained on 100% of data for another 50 epochs. Random Init. (cold-start) are models trained on 100% of the data from scratch. The dotted black line at 50 epochs represents the end of the first mega-batch training. The region between the two dotted lines shows the computationally resource intensive nature of the cold-start training to obtain optimal performance compared to warm-start and LURE. In practical settings where the training budget is limited, it is natural to maintain a single model that is updated with the latest data at regular intervals. Training a model from scratch at the onset of new data is resource-intensive and drains the training budget. Also, in a data privacy or memory-limited scenario where it is difficult to store the previously trained data, it is less intuitive and seems wasteful to sacrifice all previous computations (past knowledge acquired) for much-needed generalization. While the warm-start training damages generalization, our proposed method (LURE) brings discernible benefits in practical online settings: (1) LURE achieves faster convergence when compared to the cold-start (Random. Init.) model in settings where anytime performance is required, thus saving computational resources. (2) Improves generalization and robustness in anytime settings compared to warm-start (BL) and cold-start (Random. Init.) models. (3) LURE can also improve generalization in no-replay and buffered-replay scenarios. LURE with selective forgetting balances the trade-off between a lack of generalization in warm-start models and the enormous computational expense of retraining models from scratch through reinitialization.

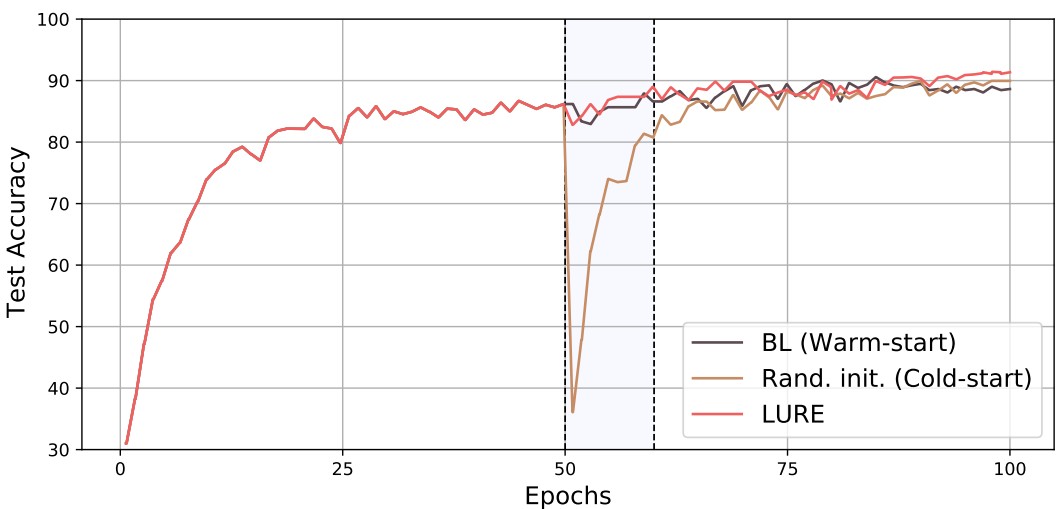

Figure 11: A comparison between ResNet models trained using warm start, random initialization, and LURE on CIFAR-10 (($|S_B| = 2$). For the first 50 epochs, the models are trained with 50% of data. Then, it is trained on 100% of the data for another 50 epochs. Random Init. (cold-start) are models trained on 100% of the data from the start. The dotted black line at 50 epochs represents the end of the first mega-batch of training. The region between two dotted lines shows the computationally resource intensive nature of the cold-start training to convergence compared to warm-start and LURE. LURE mitigates the trade-off between warm-start and cold-start training in Anytime learning settings.

