# OpenReview forum: "Learn, Unlearn and Relearn: An Online Learning Paradigm for Deep Neural Networks"
_TMLR — Accepted by TMLR_

### Review · Reviewer_aSpm · 2022-11-01

**Summary Of Contributions:**

In this paper, the authors propose Learn, Unlearn, and Relearn (LURE) for online learning with deep neural networks. The main idea is to leverage selective forgetting to improve the generalization of DNNs in online settings. The Unlearn stage selectively forgets the connections that are less relevant for the current mega-batch and retains those that are specific for the current mega-batch. The parameters are reinitialized and trained on the consecutive mega-batch based on the selection criteria. The authors conduct experiments on short sequences and long sequences to show the effectiveness of the methods.

**Audience:**

Yes

**Broader Impact Concerns:**

No concerns.

**Claims And Evidence:**

No

**Requested Changes:**

The authors should provide better understanding of the method to support the statements made in the paper. One way is to visualize the connections retained and the other way is to evaluate the robustness of the connection selection in terms of training setups.

**Strengths And Weaknesses:**

Strengths:

1. The motivation of the paper is clear.
2. The paper is well-written and the authors conduct extensive experiments to show the effectiveness of the method.


Weaknesses:

1. It would be interesting to visualize the connections retained in each mega-batch to understand how the selection method works.
2. It is not clear why the method can achieve better results than baselines. The authors argue that "selectively forgetting
extraneous information and relearning it through weight reinitialization brings discernible benefits to the model in online settings.", but it is not clear what does it mean by "extraneous information". Although the test accuracy will not change as shown by the authors, will the "extraneous information" be changed due to different training hyperparameters such as learning rates?

---

> ### Author Response · Authors · 2022-12-15
> **Reply to the Reviewer aSpm (1/2)**
>
> We thank the reviewer for the time and constructive feedback.
>
> > It is not clear why the method can achieve better results than baselines. The authors argue that "selectively forgetting extraneous information and relearning it through weight reinitialization brings discernible benefits to the model in online settings.", but it is not clear what does it mean by "extraneous information". Although the test accuracy will not change as shown by the authors, will the "extraneous information" be changed due to different training hyperparameters such as learning rates?
>
> Our proposed LURE framework is motivated by neuroscience studies on how neurogenesis-based active forgetting mediates generalization in the human brain. From a biological perspective, the growing evidence in neuroscience suggests that the process of active forgetting is a non-random process that is achieved through the selective extinction of synaptic connections [1]. Thus, active forgetting regulates the learning-memory process and prevents humans from overfitting to experiences [2, 3].
> We emulate this aspect of selective forgetting in DNNs to improve generalization. LURE interchanges between the unlearning phase, which selectively forgets the undesirable information in the model through weight reinitialization in a data-dependent manner, and the relearning phase, which emphasizes learning generalizable features. Studies based on the lottery ticket hypothesis [4] and neural network pruning [5] have confirmed that there exists a subnetwork within a densely over-parametrized network that, when trained, can achieve the same test accuracy as the original dense network. This reaffirms that most of the information pertaining to the input data can be accessed from the subnetworks, while the other connections in the network are considered to contain non-relevant information. Therefore, in line with the recent findings, we employ the SNIP algorithm [5]  from the pruning literature to identify the subnetwork post-training. SNIP uses a saliency criterion based on connection sensitivity to identify structurally important connections in the network for the given data. Therefore Our proposed model, during the unlearning phase, retains the most important connections and selectively forgets the rest of the connections that contribute least to the loss function/final performance during training (extraneous information) through weight reinitialization.
>
> This is also reflected in our empirical experiment (please refer to Appendix Section 7, Table 10). We evaluate the redundancy of the unlearned connections in the model in terms of performance and robustness. For this, we measure the performance and robustness of the full model before and after selective forgetting on clean test data and the test data subjected to different types of natural corruptions respectively. We found that the drop in test accuracy between the dense model and the sparse model (contains 20% fewer parameters than the dense model) is just 0.31% which is insignificant. The relative drop in test and robust accuracy with respect to train accuracy is less with the sparse model when compared to the dense model. Also, in robustness and generalization analysis, the sparse model either outperforms or achieves the same accuracy when compared to the dense model. This shows that the connections that are constantly reinitialized during the training contain trivial information that is indeed redundant and adds insignificant value to the model training in terms of generalization and performance. Thus, by alternating between selective forgetting (unlearning phase) and relearning, the model learn generalizable features. Thus, we empirically show that selective forgetting is crucial to retain previous information while freeing the model’s capacity to learn incoming data, thereby, improving generalization in anytime learning.
>
> We also agree with the reviewer that the term "extraneous information" can be confusing and lack clarity. We will therefore modify the manuscript accordingly.
>
> References:
> 1. Yasuda, M., Johnson-Venkatesh, E.M., Zhang, H., Parent, J.M., Sutton, M.A. and Umemori, H., 2011. Multiple forms of activity-dependent competition refine hippocampal circuits in vivo. Neuron, 70(6), pp.1128-1142.
> 2. Shuntaro Izawa, Srikanta Chowdhury, Toh Miyazaki, Yasutaka Mukai, Daisuke Ono, Ryo Inoue, Yu Ohmura, Hiroyuki Mizoguchi, Kazuhiro Kimura, Mitsuhiro Yoshioka, et al. Rem sleep–active mch neurons are involved in forgetting hippocampus-dependent memories. Science, 365(6459):1308–1313, 2019.
> 3. Lauren Gravitz. The forgotten part of memory. Nature, 571(7766):S12–S12, 2019.
> 4. Frankle, J. and Carbin, M., 2018. The lottery ticket hypothesis: Finding sparse, trainable neural networks. arXiv preprint arXiv:1803.03635.
> 5. Namhoon Lee, Thalaiyasingam Ajanthan, and Philip HS Torr. Snip: Single-shot network pruning based on connection sensitivity. arXiv preprint arXiv:1810.02340, 2018.

---

> ### Author Response · Authors · 2022-12-15
> **Reply to the Reviewer aSpm (2/2)**
>
> > The authors should provide better understanding of the method to support the statements made in the paper. One way is to visualize the connections retained and the other way is to evaluate the robustness of the connection selection in terms of training setups.
>
> We have added additional experiments in the Appendix section A6 to show the robustness of connection selection in terms of training steps. We conduct this analysis on the short sequence anytime learning scenario (full replay) on CIFAR10 with ResNet18. We globally retain 80\%  and  reinitialize (unlearn) 20\% of the parameters iteratively at the end of each mega batch of training. To study the consistency of the retained connections, we save the connection sensitivity mask (M) containing ones and zeros for the parameters retained and to be reinitialized respectively. Visualizing the connection sensitivity mask can be challenging as the parameter count in each layer of the backbone is overwhelming. Therefore, we calculate the percentage overlap of retained parameters between the connection sensitivity mask generated at the end of consecutive mega batch training as a metric for analyzing the connection robustness. Figure 4 (please refer to Section 6.4) shows the layer-wise percentage overlap of the retained parameters across consecutive mega batches. The overlap percentage of retained connections is quite high in the earlier layers across all mega batches while it decreases in the later layers (layer 4 in ResNet) as it learns class specific information pertaining to the new (unseen) samples. Although overlap percentage is less in the later layers, our method selectively unlearns few parameters in the earlier layers as well depending on the incoming data. This flexible nature of our method to unlearn and regulate connections both in the later and early layers facilitates improved generalization in anytime settings. This result is not only consistent in the full replay but also in buffered and no-replay scenarios of anytime learning. Results for the no-replay and buffered replay scenarios are provided in Figures 9a and 9b respectively.
>
> Similar experiments have been performed to analyze the robustness of connection selection to different hyperparameters, we performed comparison of the layer-wise overlap mask between the two models trained with different learning rates (Please refer to Appendix A6, Figure 10).  Results in Figure 10 demonstrate that the connection selection is more robust to the model trained under different learning rates. While most of the connections retained in the early layers remain consistent across training setups, the later layers (layer 4) showed less overlap for the model trained with different learning rates. Nevertheless, our connection selection is robust to the choice of learning rate. Therefore, the final performance (test accuracy) with our method remains unaffected to change in hyperparameters (As demonstrated in Section 6.4).

---

### Review · Reviewer_ky58 · 2022-11-20

**Summary Of Contributions:**

The paper introduces a training method for datasets that arrives in chunks.
Instead of training the network from scratch on the entire dataset whenever a new chunk arrives or fine-tuning the previous model according to the new chunk of data, they propose to employ a forgetting-based algorithm. Given the new chunk of data, they suggest reinitializing only part of the neurons in the previous model, based on their usefulness in the classification of the previous chunk. After the re-initialization, the only train part of the model.

The paper show experiments in a large array of tasks, suggesting it can improve previous baselines in several different aspects, including performance, calibration, robustness, and more.

**Audience:**

Yes

**Broader Impact Concerns:**

not relevant

**Claims And Evidence:**

No

**Requested Changes:**

- Add to all the experiments 2 baselines: training at each iteration from scratch, and fine-tuning at each iteration. This point is crucial, as it is unclear whether the current baselines are sufficient to convey the point the experiments are trying to show.

- Add a comparison of the running time of each of the methods, with emphasis on the 2 baselines suggested above. From the implementation details alone, it is unclear to me whether the proposed method can be fast in practice, or is it impractical. This point is also crucial.

- Magic numbers: There are several magic numbers along the paper (for example, sampling 0.2 of M_i, or dividing CIFAR-10 by 8 parts. Please explain for each number whether it is critical for your method to work well, or if your method is robust to it. This point can strengthen the paper, but not as critical as the points above.

- Biological motivation: In the current state, the importance of biological motivation in the paper is highly overstated -- as you don't have any experiments or clear support for such claims. The paper can be improved by tuning down these claims.

- Making sections 1-4 more concise. (can improve the paper, but not crucial)

- Elaborating on experiments done in section 5. (important point)

---------------------------
Some minor changes:

- Some of the citations seem to be misplaced: for example, when introducing "online learning", the relevant citation should be to an early work that introduces the concepts of online learning, and not Ash & Adams (2020).

- In page 3, should "0.2 X M_i" be "0.2 X |M_i|"?

- In (4), should it be f_\theta^i


**Strengths And Weaknesses:**

Strengths:

- The problem the paper tackles is interesting, and as far as I know, the given solution is novel.
- The paper is easy to follow and understand, proposing a relatively intuitive algorithm.
- The experimental design includes many different tasks, showing the advantages of the suggested algorithm across many different problems.

Weaknesses:
- The biological motivation of the work is not well supported and is based upon several superficial similarities between neural networks and the brain. Yet, it is overstated in the paper as a major contribution, which I find misleading and problematic.
While the model is indeed inspired by biological ideas, it can not serve as a major contribution, as the connection is far from being fully supported by evidence, but rather anecdotical.

- One of the main motivations of the paper, as stated repeatedly in the abstract and introduction, is to improve over 2 "naïve" strategies for learning stream-like data. The first baseline is training at each mega batch from scratch on all previously given data. The second baseline is to fine-tune the last network for the currently given data. It is stated that the first baseline is very slow, while the second baseline generalizes badly. As such, I would expect that the given method is compared to these baselines as well, as well to other methods in the literature. The comparisons should include the computation times of each method, hopefully showing that the suggested method is indeed faster than the first baseline, and generalize better than the second baseline.

- Until section 4, the paper feels very repetitive: all the ideas appear multiple times, in different variations. The entire representation can be much more concise, which will strengthen the work. On the other hand, in sections 5 and 6, many experimental designs are poorly explained, making the exact experiments that were done rather confusing. For example,  In section 5.4, many details are missing. What is the exact few-shot learning framework that is used here? It is unclear from the text itself.

- The results of the paper should be understandable without previously reading Caccia et al 2021. As many of the experimental frameworks follow the design in Caccia et al, a better explanation of each experimental design method is required. This point is amplified as Caccia et al 2021 is relatively new and not yet peer-reviewed, so without proper background reading, one cannot be sure that the experimental design is a valid one.

---

> ### Author Response · Authors · 2022-12-15
> **Reply to Reviewer ky58 (1/2)**
>
> Thank you for your extensive review. We address all your concerns point-wise.
>
> > Add to all the experiments 2 baselines: training at each iteration from scratch, and fine-tuning at each iteration. This point is crucial, as it is unclear whether the current baselines are sufficient to convey the point the experiments are trying to show.
>
> We have updated our paper with the requested baselines in both performance and robustness experiments. Please refer to Sections 5 and 6 in the paper. Warm-start models are denoted as "Baseline" whereas cold start is denoted as "Random Init" that are trained from scratch with the advent of incoming data (cold-start). Our proposed method LURE in all the settings improves generalization and outperforms both the baselines in robustness and performance. Moreover, we also observe that the performance of the cold-start models (Random Init.) is below par in low buffer and no-reply settings compared to full-replay setting (Table 1) as the model does not have access to past knowledge or data.
>
> > Add a comparison of the running time of each of the methods, with emphasis on the 2 baselines suggested above. From the implementation details alone, it is unclear to me whether the proposed method can be fast in practice, or is it impractical. This point is also crucial.
>
> The computational efficiency which we mention in the paper is regarding the epoch to reach optimal performance. As discussed in Appendix A.8, LURE achieves optimal performance early than the other baseline methods (please refer to Figure 11). Also, in practical settings where training budget is limited, it is natural to maintain a single model updated with the latest data at regular intervals. Training a model from scratch on the onset of new data is resource intensive and drains the training budget. Also, in a data privacy or memory limited scenario where it is difficult to store the previously trained data, it is less intuitive and seems wasteful to sacrifice all previous computations (past knowledge acquired) for much needed generalization. While the warm-start training damages generalization, our proposed method (LURE) brings discernible benefits in practical online settings:
> - LURE achieves optimal performance faster when compared to cold-start (Random. Init.) model in a settings where anytime performance is required, thus saving the computational resources. It is because ,similar to warm-start , LURE leverages knowledge (features, weights, etc.) from previously trained models without damaging generalization. It makes it faster than training neural networks from scratch (cold-start).
> -  Improves generalization and robustness in anytime settings compared to warm-start (BL) and cold-start (Random. Init.) model.
> - LURE can also improve generalization in no-replay and buffered-replay scenarios. Thus, LURE with selective forgetting balances the trade-off between a lack of generalization in warm-start models and saves the extra computation of retraining models from scratch through reinitialization.
>
> >  Magic numbers: There are several magic numbers along the paper (for example, sampling 0.2 of M_i, or dividing CIFAR-10 by 8 parts. Please explain for each number whether it is critical for your method to work well, or if your method is robust to it. This point can strengthen the paper, but not as critical as the points above.
>
> Thank you for pointing it out. The manuscript has been updated with the details regarding these numbers. For example, we use a 20\% of data for identifying the important connections.However, our method is robust to these parameters. We have conducted an ablation experiment (Please refer to Appendix A2, Table 9) where we consider the number of samples used for Importance estimation as low as 128. We see that the drop in test performance considering 20% of the data and 128 samples is insignificant.
>
> > Elaborating on experiments done in section 5.
>
> Details regarding the experiments are included in the paper. Please refer to Appendix section (A5) and Experimental set-up (Section 4) for a detailed setup.
>
> > This point is amplified as Caccia et al 2021 is relatively new and not yet peer-reviewed, so without proper background reading, one cannot be sure that the experimental design is a valid one.
>
> We would like to bring it to the reviewer’s notice that the paper Caccia et al 2021 titled [On Anytimelearning at Macroscale](https://proceedings.mlr.press/v199/caccia22a.html) has been peer-reviewed and accepted at a recent conference on Lifelong agents (CoLLAs 2022).

---

> ### Author Response · Authors · 2022-12-15
> **Reply to Reviewer ky58 (2/2)**
>
> > Biological motivation: In the current state, the importance of biological motivation in the paper is highly overstated -- as you don't have any experiments or clear support for such claims. The paper can be improved by tuning down these claims.
> Making sections 1-4 more concise.
>
> Our proposed LURE framework is motivated by neuroscience studies on how neurogenesis-based active forgetting mediates generalization in the human brain. From a biological perspective, the growing evidence in neuroscience suggests that the process of active forgetting is a non-random process that is achieved through the selective extinction of synaptic connections [1]. In this way, active forgetting regulates the learning-memory process and prevents humans from overfitting to experiences [2, 3].
> We emulate this aspect of selective forgetting in DNNs to improve generalization. LURE interchanges between the unlearning phase, which selectively forgets the information in the model through weight reinitialization, and relearns it, which emphasizes learning generalizable features. Based on the reviewer's suggestion  we have toned down the biological motivation and made sections 1 to 3 more concise to improve the readability of the paper.
>
> References:
> 1. Yasuda, M., Johnson-Venkatesh, E.M., Zhang, H., Parent, J.M., Sutton, M.A. and Umemori, H., 2011. Multiple forms of activity-dependent competition refine hippocampal circuits in vivo. Neuron, 70(6), pp.1128-1142.
> 2. Shuntaro Izawa, Srikanta Chowdhury, Toh Miyazaki, Yasutaka Mukai, Daisuke Ono, Ryo Inoue, Yu Ohmura, Hiroyuki Mizoguchi, Kazuhiro Kimura, Mitsuhiro Yoshioka, et al. Rem sleep–active mch neurons are involved in forgetting hippocampus-dependent memories. Science, 365(6459):1308–1313, 2019.
> 3. Lauren Gravitz. The forgotten part of memory. Nature, 571(7766):S12–S12, 2019.

---

### Review · Reviewer_8nNS · 2022-12-02

**Summary Of Contributions:**

This work introduces an online learning algorithm LURE incorporating re-initialization and re-training strategies to improve the performance of DNNs in online learning settings. Simply, LURE aims at forgetting the so-called "undesirable" parameters in a DNN and re-trains these parts thru re-initialization and re-training. Authors also brought bag of tricks for re-initialization (Unlearn). Some of Unlearn strategies here have already existed, such as SPIN. The most interesting part is the estimation of connection sensitivity and the way to compare sensitivities to decide the parts of parameters for Unlearn. Comprehensive experiments have been carried out to validate the performance of LURE. Afterall, the comparisons against baselines of other online DNN training algorithm demonstrates the superiority of LURE.

**Audience:**

Yes

**Claims And Evidence:**

Yes

**Requested Changes:**

Please address the weakness points in the above section.

**Strengths And Weaknesses:**

S1: The work is well-motivated, as re-initialzation strategies with forgetting effects could really improve the generalization performance.

S2: The manuscript is well-written, I really enjoy reading Figure 1 which illustrates the idea well.

S3: The proposed method to select parts of parameters for Unlearn operation sounds interesting (though it was not totally new), where the comparison between sailency vectors looks cool though it might be compuationally intensive.

W1: Authors mentioned RIFLE, which also selects part of parameters (FC-layer) for random re-initialization, I am wondering whether it is possible to compare RIFLE in online learning settings.

W2: There might exist some other work in transfer learning also challenges the forgetting issues while fine-tuning a model to a new set of samples. They needs to select part of parameters to freeze the weights while training the rest. Some of them are also based on attention or sailency map, such as DELTA (Li et al. 2019) at ICLR that leverages the activation maps to determine the importance of filters/neurons. I am wondering whether it is possible to use such type of sailency vector/map to Unlearn. Because, at least, they are with lower computational complexity.

W3: There are many work to analyze the gradient descent in online learning setting (Langford et al. 2009) or one-pass SGD (Zhu et al. 2021), of-course, for linear regression or two-layer NNs. I am wondering whether it is possible to extend these analyses to online learning for DNNs, through some proxy models such as Neural Tangent Kernel or two-layer NNs, and still could explain why LURE works.

Reference

Li, Xingjian, Haoyi Xiong, Hanchao Wang, Yuxuan Rao, Liping Liu, and Jun Huan. "DELTA: DEEP LEARNING TRANSFER USING FEATURE MAP WITH ATTENTION FOR CONVOLUTIONAL NETWORKS." In International Conference on Learning Representations. 2019.

Langford, John, Lihong Li, and Tong Zhang. "Sparse Online Learning via Truncated Gradient." Journal of Machine Learning Research 10, no. 3 (2009).

Zhu, Hanjing, and Jiaming Xu. "One-pass Stochastic Gradient Descent in overparametrized two-layer neural networks." In International Conference on Artificial Intelligence and Statistics, pp. 3673-3681. PMLR, 2021.

---

> ### Author Response · Authors · 2022-12-15
> **Reply to Reviewer 8nNS**
>
> We thank the reviewers for their thoughtful feedback.
>
> > W1:Authors mentioned RIFLE, which also selects part of parameters for random re-initialization, I am wondering whether it is possible to compare RIFLE in online learning settings.
>
> We thank the reviewer for pointing that out. We have updated our paper (please refer to Section 5 Results) with new experiments comparing RIFLE to our proposed method (LURE). Our method LURE outperforms RIFLE in all replay scenarios (full-replay, buffered replay, and no replay). Moreover, our method is more robust than RIFLE and all other considered baselines against natural corruption, Adversarial attacks, and noisy label training.
>
> > W2: There might exist some other work in transfer learning also challenges the forgetting issues while fine-tuning a model to a new set of samples. They needs to select part of parameters to freeze the weights while training the rest. Some of them are also based on attention or sailency map, such as DELTA (Li et al. 2019) at ICLR that leverages the activation maps to determine the importance of filters/neurons. I am wondering whether it is possible to use such type of sailency vector/map to Unlearn. Because, at least, they are with lower computational complexity.
>
> Thank you for your suggestion. DELTA (DEep Learning Transfer using Feature Map with Attention) proposes a regularized transfer learning framework that preserves the outer layer outputs of the target network rather than constraining the weights of the source network. Specifically,  DELTA aligns the outer layer outputs of two networks (Teacher network - pretrained on a large dataset, Student network - training on a small target dataset) and, through constraining a subset of feature maps that are precisely selected by the attention that has been learned in a supervised learning manner. In our method, we first measure the importance or sensitivity of each connection independently of its weight for the given task in a data-dependent manner. Then, the network selectively forgets those redundant connections and relearns them in subsequent mega-batch training. Our method works directly on the parameter and not on the activations.
>
> Though both papers address the larger problem of overfitting, the adoption of these practices from transfer learning is challenging as we do not have a teacher network that is trained on a large dataset. In our case, data arrives as a stream. Constraining the activations/feature maps between the models trained on the previous mega-batch (considering it a teacher network) might be detrimental and even end up in a  sub-optimal solution as the previous model is trained only on fewer data points compared to the current mega-batch training (online learning). However, it will be interesting to explore further tailoring those practices from transfer learning to online learning. We have addressed this directly in the manuscript as a part of future work.
>
> > W3: There are many work to analyze the gradient descent in online learning setting (Langford et al. 2009) or one-pass SGD (Zhu et al. 2021), of-course, for linear regression or two-layer NNs. I am wondering whether it is possible to extend these analyses to online learning for DNNs, through some proxy models such as Neural Tangent Kernel or two-layer NNs, and still could explain why LURE works.
>
> We thank the reviewer for suggesting these works. Empirically, we evaluated the redundancy of the unlearned connections in the model in terms of performance and robustness. For this, we measured the performance and robustness of the full model before and after selective forgetting on clean test data and the test data subjected to different types of natural corruptions respectively. We found that the drop in test accuracy between the dense model and the sparse model (which contains 20% fewer parameters than the dense model) is just 0.31% which is insignificant. The relative drop in test and robust accuracy with respect to train accuracy is less with the sparse model compared to the dense model. Also, in robustness and generalization analysis, the sparse model either outperforms or achieves the same accuracy when compared to the dense model. This shows that the connections that are constantly reinitialized during the training contain trivial information that is indeed redundant and adds insignificant value to the model training in terms of generalization and performance. Please refer to Appendix Section 7, Table 10 for more details. We believe that extending these analyses to online learning for DNNs, through Neural Tangent Kernel or two-layer NNs can be valuable for the adoption of these forgetting mechanisms for training DNNs in both online and standard settings.  In general, theoretical studies on why reinitialization during standard training improves generalization are not well explored. However, this can be done as a separate study in the future. We have included this as one of the future directions.

---

### Author Response · Authors · 2022-12-15
**General comment**

We thank all the reviewers for their thoughtful feedback. We are glad that they found our motivation, and idea, to be strong, clear, and novel and our extensive experiments insightful. Based on all reviewers' feedback, the following modifications have been made to the manuscript which is in blue color:

**First three sections:**

- The paper is improved by tuning down the biological motivation in the claims and concising the first three sections.
- Minor corrections were rectified.
- Added explanations for some of the hyperparameters used in the proposed methodology.

**Experimental set-up:**

-  Added the details about the Anytime setup used for experiments in the result section.
- Added the training details and hyperparameters used for training the LURE model and its baselines.

**Results:**
-  Updated all the Figures and Tables in the Result Sections 5 and 6 ( Table 1, Table 2, Table 3, Table 4, Table 5, Table 6, Figs 2 and 3) with three important baselines - BL (Warm-start model), Random Init. ( Cold-start model) and RIFLE. Our method (LURE) outperforms the baselines in all three replay scenarios (full-replay, buffered replay, and no replay). Moreover, our method (LURE) is more robust compared to other baselines against natural corruption, Adversarial attacks, and noisy label training.
- Added more details regarding the few-shot experimental setup.
- Added more information and comparison with the baselines in terms of computational efficiency in the number of epochs to optimal performance and the advantages it brings in practical settings (A.8).
- Due to space constraints, we have added more information requested by reviewers in the Appendix. Experiments such as convergence to flatter minima, and model calibration has been shifted to Appendix (A.3 and A.4).

**Additional experiments:**
- Section 6.4 - Robustness of the connection selection across training steps.
- Appendix A.2 - Varying the quantity of data used for Importance estimation.
- Appendix A.5 - Implementation details and hyperparameters.
- Appendix A.6 - Robustness of the connection selection to change in learning rate.
- Appendix A.7 - Evaluating the redundancy of parameters removed during the unlearning phase.
- Appendix A.8 - Comparison of LURE with warm-start and cold-start training in terms of computation.

*Please let us know in case we have missed any of your feedback.*

---

### Decision · Action_Editors · 2023-01-21

**Recommendation:** Accept as is

**Comment:**

The paper explores an interesting approach to online learning that makes the use of selective forgetting. The results show favourable results compared to re-training, a naive warm start, as well as other approaches in the literature.

Concerns of the reviewers mainly focussed on lack of clarity, missing baselines, and slightly unconvincing biological motivation. With additional experimental details, baselines, and some toning down of the bio-inspired motivation, the paper meets TMLR's acceptance criteria.

Congratulations on the acceptance to TMLR!

**Audience:**

The paper concerns online learning, a popular area of study in deep learning, and proposes a promising approach which demonstrates desirable properties. This paper will likely be of interest to some of TMLR's audience.

**Claims And Evidence:**

The claims are supported by convincing evidence. The authors provided additional baselines and an extensive analysis in the appendix, which satisfied the concerns of the reviewers.

I have one minor request to the authors:  In the introduction please replace "We demonstrate the efficacy of LURE in multiple architectures" with "We demonstrate the efficacy of LURE in multiple _convolutional_ architectures". These days, there are popular architectures that don't follow the classical CNN design, and this change would clarify that the empirical results here are restricted to various CNNs.